# Human Heme Oxygenase-1 Induced by Interleukin-6 via JAK/STAT3 Pathways Is a Tumor Suppressor Gene in Hepatoma Cells

**DOI:** 10.3390/antiox9030251

**Published:** 2020-03-19

**Authors:** Kun-Chun Chiang, Kang-Shuo Chang, Shu-Yuan Hsu, Hsin-Ching Sung, Tsui-Hsia Feng, Mei Chao, Horng-Heng Juang

**Affiliations:** 1Department of General Surgery, Min-Sheng General Hospital, Tao-Yuan 33302, Taiwan; robertviolet6292@gmail.com; 2Department of Anatomy, College of Medicine, Chang Gung University, Kwei-Shan, Tao-Yuan 33302, Taiwan; D0501301@stmail.cgu.edu.tw (K.-S.C.); hsusy@mail.cgu.edu.tw (S.-Y.H.); hcs@mail.cgu.edu.tw (H.-C.S.); 3Institute of Medicine Science, College of Medicine, ChSang Gung University, Kwei-Shan, Tao-Yuan 33302, Taiwan; 4School of Nursing, College of Medicine, Chang Gung University, Kwei-Shan, Tao-Yuan 33302, Taiwan; thf@mail.cgu.edu.tw; 5Department of Microbiology and Immunology, College of Medicine, Chang Gung University, Kwei-Shan, Tao-Yuan 33302, Taiwan; 6Department of Hepato-Gastroenterology, Liver Research Center, Chang Gung Memorial Hospital-Linkou, Kwei-Shan, Tao-Yuan 33302, Taiwan; 7Department of Urology, Chang Gung Memorial Hospital-Linkou, Kwei-Shan, Tao-Yuan 33302, Taiwan

**Keywords:** HO-1, IL-6, ROS, STAT3, JAK, luteolin, AG490, HepG2, Hep3B

## Abstract

Heme oxygenase-1 (HO-1) has several important roles in hepatocytes in terms of anti-inflammation, anti-apoptosis, and antioxidant properties. Interleukin-6 (IL-6) is a pleiotropic cytokine associated with liver regeneration and protection against injury. The aim of this study was to determine the potential crosstalk between HO-1 and IL-6, and to elucidate the signaling pathways involved in the induction of HO-1 by IL-6 in human hepatoma cells. Ectopic overexpression of HO-1 not only attenuated cell proliferation in vitro and in vivo, but also blocked the reactive oxygen species (ROS) induced by H_2_O_2_ and the pyocyanin in HepG2 or Hep3B cells. IL-6 expression was negatively regulated by HO-1, while IL-6 induced signal transducer and activator of transcription 3 (STAT3) phosphorylation and HO-1 gene expression in HepG2 cells. The co-transfected HO-1 reporter vector and a protein inhibitor of the activated STAT3 (PIAS3) expression vector blocked the IL-6-induced HO-1 reporter activity. Both interferon γ and interleukin-1β treatments induced STAT1 but not STAT3 phosphorylation, which had no effects on the HO-1 expression. Treatments of AG490 and luteolin blocked the JAK/STAT3 signaling pathways which attenuated IL-6 activation on the HO-1 expression. Our results indicated that HO-1 is the antitumor gene induced by IL-6 through the IL-6/JAK/STAT3 pathways; moreover, a feedback circuit may exist between IL-6 and HO-1 in hepatoma cells.

## 1. Introduction

Heme oxygenase-1 (HO-1) is an inducible enzyme that can degrade toxic heme into carbon monoxide (CO), ferrous iron, and biliverdin [1]. In general, the level of HO-1 expression is low when cells are at the resting stage, but profoundly enhanced by certain conditions or stimulators, such as heavy metals, heme, cytokines, hormones, lipid metabolites, and oxygen status [2]. In addition, recent studies have shown that HO-1 is involved in the improvements of several events, such oxidation stress, airway inflammation, functions of adipocytes, obesity, steatohepatitis, atherosclerosis, diabetes mellitus, brain hemorrhage, as well as neuroprotection [3,4,5,6,7,8]. This is especially essential in the health maintenance of the liver which contains about 15% of body heme [9,10,11,12]. Numerous studies have proved that the anti-inflammatory, antioxidant, pro-angiogenic, anti-thrombotic, and anti-apoptotic effects of HO-1 are derived from its catabolic byproducts, CO and biliverdin, which render HO-1 as a tissue protector and a good candidate for investigating new therapeutic interventions [1,8,13]. However, several studies have extensively demonstrated the positive role of HO-1 in cancer progress [14,15,16].

Interleukin 6 (IL-6) is a pleiotropic cytokine exerting multiple biologic functions. Different IL-6 pathways are involved in the physiology and pathophysiology of the liver and they are crucial for the development of hepatocellular carcinoma [17]. IL-6 is able to activate acute phase proteins in response to certain inflammatory conditions or infective diseases, and is important for hepatocyte homeostasis and liver regeneration after liver damage through the activation of IL-6/JAK/STAT3 pathways [17,18].

Previous reports remain inconclusive in terms of the regulation between IL-6 and HO-1. For example, studies have indicated different cell types in which HO-1 can be one of the IL-6 positive or negative responding genes through the modulation of STAT3 phosphorylation [10,19,20,21]. A further study revealed that HO-1 attenuates IL-6-induced STAT3 phosphorylation in human keratinocyte HaCaT cells [22]; however, high levels of HO-1 expression stimulated autocrine IL-6 production in human multiple myeloma U226 cells [23]. Although early studies have suggested that IL-6 induced rat HO-1 gene expression through the JAK/STAT pathway in hepatocytes [24] and the induction of IL-6 on human HO-1 gene expression in the human hepatoma cells during acute-phase reaction [25], the potential crosstalk between IL-6 and HO-1 has not yet been studied. In this study, we aim to elucidate the function of HO-1 on human hepatoma cells and to investigate molecular mechanisms whereby IL-6 modulates human HO-1 gene expression.

## 2. Material and Methods

### 2.1. Cell Culture and Chemicals

HepG2 and Hep3B cell lines were obtained from the Bioresource Collection and Research Center (BCRC, Hsinchu, Taiwan) and maintained in culture medium RPMI 1640 (Life Technologies, Rockville, MD, USA) with 10% fetal calf serum (FCS; HyClone, Logan, UT, USA). The AG490, a selective inhibitor of JAK/STAT3 activation, and luteolin were from Sigma (St. Louis, MO, USA). The anti-human IL-6 monoclonal antibody (MAB2061) was from R & D Systems, Inc. (Minneapolis, MN, USA). IL-6, INFγ, and IL-1β were from PeproTeck (Rehovot, Israel).

### 2.2. Expression Vectors and Overexpression

Full-length human IL-6, STAT3, and PIAS3 expression vectors were constructed as previously described [26,27]. Briefly, the human STAT3 (pcDNA-STAT3) and PIAS3 (pcDNA-PIAS3) expression vectors were constructed by cloning the STAT3 cDNA vector (MCG: 4909141) and the PIAS3 cDNA vector (MGC: 3528679), respectively, into the pcDNA3 expression vector (Invitrogen, Carlsbad, CA, USA). The human IL-6 expression vector (pcDNA-IL-6) was constructed by cloning the IL-6 cDNA vector (MCG: 9215) after digestion with Eco RI and Not I into the pcDNA3.1/Zeo expression vector (Invitrogen). The human HO-1 expression vector was constructed by cloning a full-length HO-1 cDNA (MGC:1723; Invitrogen) into the pcDNA3.1/Zeo expression vector (Invitrogen) with Eco R1 sites. The HO-1 or IL-6 expression vector was transiently transfected into HepG2 or Hep3B cells, respectively, by electroporation, as previously described [28]. Cells were maintained in RPMI medium with 10% FCS and 100 μg/mL of Zeocin (Invitrogen). HO-1 or IL-6 expression in resistant colonies (HepG2-HO1, Hep3B-HO1, and HepG2-IL-6) was evaluated by immunoblot, RT-qPCR, or ELISA, as described below. The mock-transfected HepG2 cells (HepG2–DNA) and Hep3B (Hep3B–DNA) were transfected with the control pcDNA3.1/Zeo expression vector and also selected by Zeocin.

### 2.3. Immunoblot Assay

Equal quantities of cell extracts were separated on a 10% SDS-PAGE gel, transferred and analyzed by the Western Lightning Plus-ECL detection system (Perkin Elmer, Inc., Waltham, MA, USA). Antibodies against heme oxygenase (HO-1; Stressgen, Victoria, BC, Canada), JAK, phospho-JAK, AKT, Phospho-AKT, STAT1, Phospho-STAT1, STAT3, Phospho-STAT3 (Cell Signaling, Danvers, MA, USA), and β-actin (Millipore, Temecula, CA, USA) were used. Band intensities were recorded using the Chemi Genius II BioImaging System of Syngene (Cambridge, UK) and analyzed using the GeneTool Program of ChemiGenius (Syngene).

### 2.4. Detection of ROS with Flow Cytometer and Total ROS Analysis

ROS was analyzed using the FACS-Calibur Cytometer (BD Biosciences, Franklin Lakes, NJ, USA), as previously described [15]. In brief, cells were harvested with trypsin after being cultured for 48 h. Cells were washed and 20 μL of carboxy-H_2_DCFDA was added to the cell pellet, and then cells continued to incubate for 30 min. After treatment with or without 20 μM of ROS inducer (pyocyanin) for 1 h, cells were suspended in 500 μL of PBS. Total ROS was analyzed using the ROS detection kit (Enzo Life Science, Farmingdale, NY, USA). Cells were cultured in a 96-well plate for 48 h and washed twice with PBS. Two hundred μL of 20 μM DCF-DA were added and further incubated for 30 min. The 0–1000 μM of H_2_O_2_ in RPMI 1640 medium with 10% FCS was added for another 1 h, then the intensity of DCF-DA fluorescence was quantified with the Chameleon Fluoro-Lumino-Photometer (Turku, Finland)

### 2.5. RT-qPCR

Total RNA was isolated with Trizol reagent and synthesized cDNA using the superscript III preamplification system (Invitrogen). The real-time polymerase chain reaction (PCR) was performed using a CFX Connect Real-Time PCR system (Bio-Rad Laboratories, Foster city, CA, USA), as previously described [29]. FAM dye-labeled TaqMan MGB probes and PCR primers for human IL-6 (Hs00985639_m1) and HO-1 ((Hs00157965_m1) were purchased from Applied Biosystems (Foster City, CA, USA). For the internal positive control, β-Actin (Hs01060665_g1) was used with a FAM dye-labeled TaqMan MGB probe.

### 2.6. Matrigel Invasion Assay

The invasion ability of the cells was determined through an in vitro Matrigel invasion assay. Cells that migrated to the Matrigel-coated transmembrane for 48 h were fixed with 4% paraformaldehyde and then stained with 0.1% crystal violet solution for 10 min. The quantity of cells that invaded the Matrigel was recorded microscopically (IX71, Olympus, Tokyo, Japan). The methods of analysis and counting were performed as described previously [29].

### 2.7. Cell Proliferation

The cell proliferations were measured using ^3^H-thymidine incorporation assay, as previously described [29]. Cells were cultured in each well of a six-well plate. After the required incubation time, as indicated, 1 μCi/mL of ^3^H-thymidine was added and then continued to incubate for another 4 h. Cells were washed with cold phosphate buffer saline (PBS) and then fixed with cold 5% trichloroacetic acid. Cells were lysed by adding 500 μL of 0.5 N NaOH. The solubilized cell solution (400 μL) was mixed with 1 mL of scintillation cocktail and counted in a liquid scintillation analyzer (Packard BioScience, Downers Grove, IL, USA).

### 2.8. Enzyme-Linked Immunosorbent Assay (ELISA)

Cells were incubated with 1 mL of RPMI medium with 10% FCS in a 24-well-plate (2 × 10^4^ cells/well) for a period of 36 h. After incubation, we collected supernatants from each well and measured IL-6 levels in the conditioned media by an IL-6 ELISA kit according to manufacturer’s instructions (Cat No. D6050; R & D Systems, Inc.). The IL-6 levels were adjusted by the concentrations of protein in the whole cell extract, which was measured using a BCA protein assay kit (Pierce, Rockford, IL, USA), as previously described [28].

### 2.9. Report Vector Constructs

The IL-6 reporter vector was constructed as described previously [30]. The STAT3 reporter vector (pSTAT3-TA-Luc) was purchased from Clontech Laboratories Inc. (Mountain View, CA, USA). The DNA fragment of the promoter/enhancer of the HO-1 gene was isolated from a BAC clone (CTA-286B10; a gift from the Wellcome Trust, Sanger Institute, Cambridge, UK). The BAC clone was digested with Apa I or Pst I and three DNA fragments that contained the HO-1 promoter/enhancer DNA fragment were subcloned into the pGEM11 or pGEM5 vectors, respectively. One DNA fragment was digested with Xho I and ligated to the pbGL3 reporter vector (Promega Bioscience, San Luis Obispo, CA, USA), and termed pHO1-X, which contained the 5′-flanking region (−106 to +22) of the HO-1 gene. Another DNA fragment was digested with Apa I and Xho I, and then ligated to the pHO1-X reporter vector which was termed pHO1-AX, containing the 5′ flanking region (−876 to +22) of the HO-1 gene. Another DNA fragment in the pGEM11 vector was digested with Kpn I and Apa I, and then ligated to pHO-1AX, which was termed pHO1-AK, containing the 5′ flanking region (−4333 to +22) of the HO-1 gene. The pGL3 promoter vector (pPGL3), which contained the SV40 promoter, was purchased from Promega Biosciences. Proper ligation and orientation were confirmed using extensive restriction mapping and sequencing.

### 2.10. Reporter Assays

Cells were plated onto a 24-well plate at 1 × 10^4^ cells/well for one day prior to transfection. Cells were transiently transfected using the TransFast transfection reagent (Promega Biosciences) with reporter vectors, pCMVSPORTβgal (Life Technologies), and expression vectors, as indicated, for 4 h. After another 48 h of incubation with RPMI 1640 medium with 10% FCS, the luciferase activity was determined in relative light unit using a Synergy H1 microplate reader (BioTek, Beijing, China) and was adjusted according to the β-galactosidase enzymatic activity, as described previously [30].

### 2.11. Xenograft Animal Study

These studies met the criteria of the Guide for Laboratory Animal Facilities and Care as promulgated by the Council of Agriculture Executive Yuan, Taiwan and all methods were performed in accordance with the “Animal Welfare Law and Policy” (LAW3ANI). The protocol was approved by the Chang Gung University Animal Research Committee (Permit Number: CGU15-154). Male nude mice (BALB/cAnN-Foxn1, four weeks old) were randomized into two groups: Hep3B–DNA (*n* = 6) and Hep3B-HO1 (*n* = 6). Animals were anesthetized intraperitoneally and equal volumes of cells (8× 10^6^/100 μL) were injected subcutaneously on the side of the back. Tumor volume was measured at three-day intervals using vernier calipers and calculated as π/6 × larger diameter × (smaller diameter)^2^, as described previously [31].

### 2.12. Statistical Analysis

Results are expressed as the mean ± S.E. of at least three independent experiments. Statistical significance (* *p* < 0.05; ** *p* < 0.01) was determined by a t-test and one-way ANOVA using SigmaStat software for Windows, version 2.03 (SPSS Inc, Chicago, IL, USA). The post-hoc analysis was used to correct for multiple comparisons.

## 3. Results

### 3.1. HO-1 Retards Cell Proliferation and Cell Invasion of Human Hepatoma Cells

In order to determine the biologic functions of HO-1 in hepatoma cells, we overexpressed ectopic HO-1 into the cells. Immunoblot assays confirmed that transient ectopic HO-1-overexpressed HepG2 and Hep3B cells expressed higher levels of HO-1 compared to mock-transfected cells (HepG2–DNA and Hep3B–DNA) (Figure 1A). The thymidine incorporation assays revealed that overexpression of HO-1 in HepG2 (Figure 1B) and Hep3B (Figure 1C) cells downregulated cell proliferation. Moreover, the Matrigel invasion assays indicated that HO-1 overexpression blocked 75% of cell invasion compared to HO-1 mock-transfected Hep3B cells (Figure 1D).

### 3.2. Ectopic Overexpression of HO-1 Inhibits Tumorigenesis of HepG3 Cells

We continued to determine the anti-proliferation activity of HO-1 in vivo by using xenograft animal studies. Hep3B–DNA and Hep3B-HO1 cells were injected subcutaneously into the back of nude mice to determine the effects of HO-1 on tumorigenesis. After 38 days of growth, the tumor volume of tumors derived from Hep3B–DNA cells was 2.08 times the size of those from the Hep3B-HO1 cells (274.13 ± 17.82 mm^3^ vs. 132.15 ± 18.57 mm^3^) (Figure 1E). The tumor weight of tumors derived from Hep3B–DNA cells was about 1.56 times the weight of tumors from the Hep3B-HO1 cell group (Figure 1F). The results of RT-qPCR assays confirmed that HO-1 was overexpressed, but IL-6 gene expression was downregulated in the xenograft tumors derived from Hep3B-HO1 cells (Figure 1G).

### 3.3. HO-1 Alleviates ROS Generation in Hep3B Cells

We determined the antioxidant response of HO-1 in human Hep3B cells. Results of the total ROS assays indicated that treatments of 125–1000 μM of H_2_O_2_ induced 55.2- to 61.3-fold increase in ROS immunofluorescence intensity in Hep3B–DNA cells, while the same dosage of H_2_O_2_ induced only 17.5- to 40.1-fold increase in ROS immunofluorescence intensity in Hep3B-HO1 cells compared to the vehicle treatment (Figure 2A). Further flow cytometry assays indicated that ROS generation was stimulated by 20 μM of pyocyanin, a ROS inducer, in Hep3B–DNA cells; however, ectopic HO-1 overexpression blocked the pyocyanin-stimulated ROS generation in Hep3B-HO1 cells compared with Hep3B–DNA cells (Figure 2B).

### 3.4. HO-1 Downregulates IL-6 Expression in Hep3B Cells

In order to determine the anti-inflammatory characteristics of HO-1 in hepatoma cells, we evaluated the effect of HO-1 on IL-6 gene expression. The results of the ELISA indicated that transient overexpression of HO-1 downregulated IL-6 secretion in Hep3B cells compared with mock-transfected (Hep3B–DNA) cells (Figure 3A). The quantitative results of RT-qPCR assays revealed that ectopic HO-1 overexpression enhanced 12.08 ± 1.88 fold and 0.41 ± 0.10 fold, respectively, the HO-1 and IL-6 mRNA levels in Hep3B cells, (Figure 3B) indicating that ectopic overexpression of HO-1 downregulated the gene expression of IL-6 compared with Hep3B–DNA cells. Reporter assays also showed similar results. The co-transfection of HO-1 expression vector blocked IL-6 reporter activity in a dosage-dependent manner (Figure 3C).

### 3.5. Ectopic IL-6 Overexpression or IL-6 Treatment Enhances HO-1 Expression

We investigated the crosstalk between IL-6 and HO-1 in hepatoma cells, since IL-6 is known as one of the HO-1 stimulators. The IL-6 expression vector was transient transfected into HepG2 cells by electroporation. The results of our ELISA clearly revealed that the levels of IL-6 secretion were upregulated more than 26.6-fold after ectopic IL-6 overexpression (Figure 4A). The quantitative results of RT-qPCR assays revealed that ectopic IL-6 overexpression enhanced 128.1 ± 19.9 fold and 20.1 ± 5.3 fold, respectively, the IL-6 and HO-1 mRNA levels, indicating that ectopic overexpression of IL-6 induced gene expression of HO-1 in HepG2 cells (Figure 4B). The immunoblot assays showed the similar results (Figure 4C). Further immunoblot assays indicated that IL-6 treatment induced HO-1 gene expression either time-dependently (Figure 4D, top) or dosage-dependently (Figure 4D, bottom). The IL-6 monoclonal antibody blocked the HO-1 activation under IL-6 treatment in HepG2 cells (Figure 4E).

### 3.6. IL-6 Induces HO-1 Reporter Activity

We continued to investigate the gene regulation of HO-1 by IL-6 in hepatoma cells. Figure 5A shows the map of human HO-1 reporter vectors. The reporter activities of the HO-1 reporter vectors in comparison to the reporter vectors containing with (ppGL3) or without (pbGL3) SV40 promoter, respectively, are described in Figure 5B. The results of the reporter assays implied that co-transfection of pGL3-HO1-KX reporter vectors with human IL-6 expression vectors upregulated HO-1 reporter activities (Figure 5C). The results of the 5′ deletion reporter assays suggested a putative IL-6 response element located between -876 bp and -106 bp upstream of the transcription-initiation site of the human HO-1 gene (Figure 5D).

### 3.7. IL-6-Induced HO-1 Reporter Activity Is Dependent on STAT3 Signaling

Transient overexpression of IL-6 or STAT3 expression vectors upregulated the reporter activity of the STAT3 reporter vector in HepG2 cells; moreover, the co-transfected protein inhibitor of the activated STAT3 (PIAS3) expression vector blocked the activation of STAT3 in the STAT3 reporter activity (Figure 5E). Further reporter assays showed that transient overexpression of the IL-6 expression vector enhanced 2.4-fold the HO-1 reporter activity in HepG2 cells. Notably, transfecting the STAT3 expression vector alone did not affect the HO-1 reporter activity; however, the co-transfected STAT3 expression vector enhanced the IL-6 activation in the HO-1 reporter activity. Additionally, reporter assays revealed that the transient co-transfected PIAS3 expression vector blocked the activation of IL-6 and STAT3 in HO-1 reporter activities in HepG2 cells (Figure 5F).

### 3.8. IL-6 induces HO-1 Expression via STAT3 Phosphorylation

We continued to determine the specific STAT3 signal pathways and the effect of IL-6 on HO-1 gene expression. Immunoblot assays revealed that IL-6 treatments induced STAT3 phosphorylation in HepG2 cells dose-dependently (Figure 6A). Results of reporter assays using the HO-1 reporter vector indicated that only IL-6 treatment, not IL-1β or interferon γ (INFγ), could affect HO-1 reporter activity in HepG2 cells (Figure 6B). Further immunoblot assays showed that both INFγ (Figure 6C,E) and IL-1β (Figure 6D,F) treatments induced STAT1, not STAT3, phosphorylation.

### 3.9. AG490 and Luteolin Block Activation of IL-6 on HO-1 Expression

In order to confirm that JAK-STAT3 phosphorylation is a major factor for IL-6 activation in HO-1 expression, we used AG490 to block the activation of IL-6 on STAT3 phosphorylation. Immunoblot assays showed that IL-6 induced JAK/STAT3 phosphorylation in HepG2 cells; however, 20 μM of AG490 blocked the IL-6 effect not only on JNK/STAT3 phosphorylation, but also on HO-1 protein levels (Figure 7A,C). Further immunoblot assays revealed that luteolin (20 μM) did not affect AKT activity, while it downregulated STAT3 phosphorylation, thus attenuating IL-6 activation in HO-1 expression (Figure 7B,D). Reporter assays using the pGL3-HO1-AX reporter vector also indicated that both AG490 and luteolin decreased HO-1 reporter activity induced by IL-6 (Figure 7E). Taken together, these results prove that IL-6 affects HO-1 expression through specific JAK/STAT3 signaling.

## 4. Discussion

The major function of heme oxygenase (HO) is to catalyze heme into CO, biliverdin, and iron [32,33]. HO comprises two isoforms, HO-1 and HO-2, and HO-2 is a constitutively expressive protein mainly found in the brain and testis [34]. Unlike HO-2, HO-1 is a highly stress-induced protein with a low expression the tissue. Substances that could increase cellular stress include heavy metals, endotoxin, cytokines, hypoxia, and nitric oxide, all of which have shown to upregulate HO-1 expression [35,36,37]. Since the anti-inflammatory, anti-apoptotic, and pro-angiogenetic properties of HO-1, the induction of HO-1 is widely deemed a physiological protective response against stress and adaptation to stress [38,39,40]. However, studies have found that HO-1 has a dual role in different types of cancers [41].

Investigations into the role of HO-1, with contrary results, have been extensively discussed in the study of liver carcinogenesis in vitro or in vivo [42,43,44,45,46,47,48]. Studies of immunohistochemical staining showed that HO-1 is frequently overexpressed in human hepatocellular carcinoma (HCC); however, HO-1 expression did not seem to influence the prognosis of HCC [42,43]. Studies using the rat hepatoma AH1368 cells and mouse hepatoma cells, Hepa129 and Hepa1-6, indicated that HO-1 is regarded as an oncogene in orthotopic tumor assays [42,44]. However, results from other studies showed that overexpression of HO-1 in HepG2 retarded cell proliferation, migration, invasion, and orthotopic tumor growth, which is in agreement with our in vitro or in vivo studies using human hepatoma cells [47,48]. The mechanisms of HO-1 in the attenuation of HCC progression through the microRNA pathway have been discussed [48]. Moreover, as shown in Figure 2, our results also demonstrated that the overexpression of HO-1 in Hep3B cells attenuated H_2_O_2_-induced ROS generation. These results supported recent studies, suggesting that the upregulation of HO-1 alleviated H_2_O_2_-induced ROS and oxidative injury in HepG2 cells [49,50], although, in this study, we did not determine whether ectopic HO-1 overexpression inhibited the apoptosis of hepatoma cells. Since a recent study indicated that IL-6 drives ROS reduction by increasing Nrf 2, a transcription activator of the HO-1 gene, expression in human islet β-cells [51], further investigations into the role of IL-6–HO-1 crosstalk in hepatoma cells, in terms of ROS synthesis, are warranted.

The role of HO-1 in anti-inflammation has been well-established [52]. In the present study, we found that expressions of IL-6 were downregulated in HO-1-overexpressed Hep3B cells compared to Hep3B–DNA cells (Figure 3). Results of this study are similar to other previous studies which demonstrated that HO-1 downregulated IL-6 expression in microglial BV2 cells, macrophages RAW264.7 cells, and prostate carcinoma PC-3 cells [53,54,55]. However, the effect of HO-1 on IL-6 expression might be dependent on cell types, since a study of multiple myeloma cells showed contrary results in which HO-1 induced IL-6 expression [23].

IL-6 is now a well-known cytokine with pleotropic functions in the liver [17] and has been shown to be one of the HO-1 stimulators [24]. Our results clearly show that ectopic overexpression of IL-6 or IL-6 treatments can stimulate HO-1 expression in HepG2 cells. IL-6 monoclonal antibody treatment could attenuate an IL-6 effect on HO-1 induction (Figure 4). The induction of HO-1 by IL-6 has been investigated by several cell types, including human hepatoma cells [10,23,25]. However, studies in the human skin cells indicated that IL-6 downregulated HO-1 expression in vitro and in vivo [20,21].

Since HO-1 has anti-inflammatory characteristics [52], the IL-6-induced HO-1 seems to bring a balance in preventing cellular “over inflammation” in hepatoma cells. From the results in Figure 3 and Figure 4, our study indicated that HO-1 is highly induced by IL-6 and HO-1 negatively regulated IL-6, indicating a negative feedback loop between HO-1 and IL-6. However, another study of human multiple myeloma cells showed a positive crosstalk between IL-6 and HO-1 [23]. In addition to HO-1 blocking the ROS synthesis after H_2_O_2_ or pyocyanin stimulation in human hepatoma cells (Figure 2), our data supported the belief that upregulation of HO-1 plays a pivotal role in cell adaptation to oxidative stress and cytokine insults, and the negative feedback between HO-1 and IL-6 might be partly due to the anti-inflammatory effect of HO-1 [14].

It has been implied that there are four signaling cascades involved in the HO-1 expression, including p38/MAPK, PI3K/AKT, JAK/STAT, and toll-like receptor pathways [52]. IL-6 is able to activate Janus kinases (JAK), which further phosphorylate STAT1 and STAT3 [23,49]. The phosphorylation of STATs induces the nuclear translocation of STATs, which further binds to the specific binding site within the promotor areas of gene-influencing gene expression. As shown in Figure 6A, IL-6 treatment induced STAT3 phosphorylation. To facilitate our understanding of the regulation of IL-6 on the human HO-1 gene expression, we performed a 5′ deletion reporter assay. The results in Figure 5D indicate that there is a putative IL-6 responsive element located between -876 bp and -106 bp upstream of the transcription initiation site of the human HO-1 gene, which is in agreement with an early study that suggested three IL-6 responsive elements (AGTGANGNAA) within this DNA fragment [25]. Interestingly, the transfection of the STAT3 expression vector only increased STAT3 reporter activity, but not HO-1 reporter activity (Figure 5E,F). However, co-transfection of IL-6 and STAT3 expression vectors strongly increased HO-1 reporter activity compared to the IL-6 transfection-only group or control group. Our results suggest that STAT3 needs the activation of IL-6 to induce HO-1 expression. Further reporter assays indicate that PIAS3 can block the induction of IL-6 on HO-1 expression. Collectively, our data suggests that IL-6 upregulates human HO-1 gene expression via the STAT3 signaling pathway.

Since IL-6 could activate both STAT1 and STAT3 [52], to further confirm the IL-6 effect on HO-1 being through the STAT3 pathway, IL-1β and INFγ were applied to treat the HepG2 cell. As shown in Figure 6B, neither IL-1β nor INFγ could affect HO-1 reporter activity in HepG2 cells. Further immunoblot assays showed that IL-1β and IFNγ induced STAT1 but not STAT3 phosphorylation, which did not affect the HO-1 expression (Figure 6C,D). This result supports our findings in Figure 5F and Figure 6B, which show that the IL-6-activated STAT3 signaling pathway specifically induces HO-1 expression.

We further used AG490 to treat HepG2 cells in order to block the Janus kinase 2 signaling pathway. As shown in Figure 7A, AG490 treatment decreased IL-6-induced STAT3 phosphorylation and HO-1 expression. The results agree with a previous study in human multiple myeloma cells [23]. Luteolin, a natural flavonoid with potent anti-inflammatory properties, has believed to be able to block the JAK/STAT pathway [56]. After luteolin treatment, IL-6 affected either STAT3 phosphorylation or HO-1 expression was inhibited (Figure 7B). A recent study indicated that luteolin treatment blocked STAT3 phosphorylation in hepatic stellate cells to attenuate the expression of STAT3-regulated proteins, which is in agreement with our study [57]. Similar results from the reporter assays confirmed that both AG490 and luteolin could downregulate IL-6-stimulated HO-1 expression in HepG2 cells (Figure 7). Taken together, our results suggest that IL-6 induces human HO-1 gene expression through the JAK/STAT3 signaling pathway.

## 5. Conclusions

Our results confirm that overexpression of HO-1 not only attenuates cells proliferation and invasion, but also blocks ROS induced by H_2_O_2_ in human hepatoma cells. IL-6 induces human HO-1 gene expression via the JAK/STAT3 pathway in HepG2 cells; however, there is a negative autocrine between expressions of HO-1 and IL-6 in human hepatoma cells. Our results suggest that HO-1 is an IL-6-induced anti-tumor gene in the human hepatoma cells. Further studies on the application of HO-1 and its inducers against tumorigenesis and inflammatory effects in hepatocellular carcinomas are warranted.

## Figures and Tables

**Figure 1 antioxidants-09-00251-f001:**
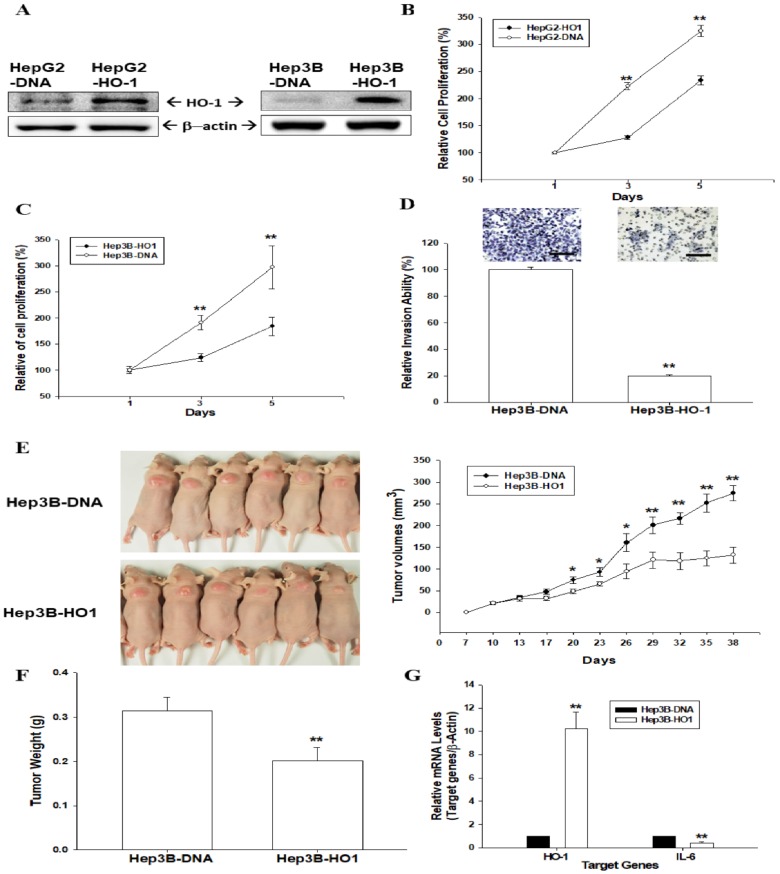
HO-1 attenuates cell proliferation and invasion of human hepatoma cells. (**A**) HepG2 (left) and HepG3 (right) cells were transiently overexpressed the HO-1 expression vector. The protein levels of HO-1 were determined by immunoblot assays. The cell proliferation rates in HepG2–DNA and HepG2-HO1 (**B**), and also in Hep3B–DNA and Hep3B-HO1 (**C**) cells, were determined by ^3^H-thymidine incorporation assays (**D**) The invasive abilities of Hep3B–DNA and HepG3B-HO-1 cells were determined by in vitro Matrigel invasion assays. Data are presented as the mean percentage (±SE, *n* = 3) of invasion ability in relation to that of the Hep3B−DNA cell group. The scale bar is 50 μm. Four-week-old male athymic nude (nu/nu) mice were randomized into two groups: Hep3B–DNA (*n* = 6) and Hep3B-HO1 (*n* = 6). Hep3B–DNA and Hep3B-HO1 cells (8 × 10^6^) were injected into the dorsal subcutaneous area, respectively. Tumor growth rates (**E**) were measured every 3 days, starting at the first week of growth (day seven) in which the tumors became perceptible under the skin after inoculation. The tumor weight (**F**) and mRNA levels of HO-1 and IL-6 (**G**) were determined by RT-qPCR after animals were sacrificed. (*, *p* < 0.05; **, *p* < 0.01).

**Figure 2 antioxidants-09-00251-f002:**
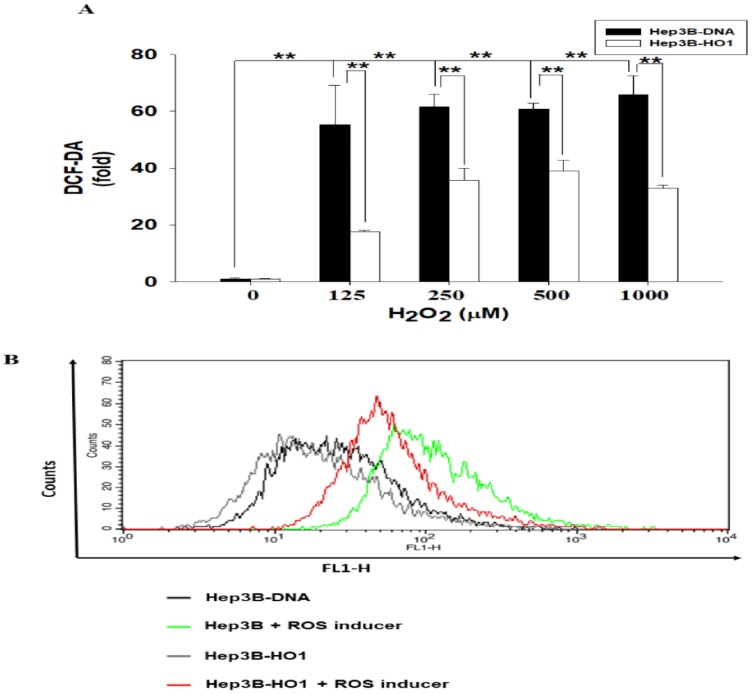
HO-1 alleviates ROS generation in human hepatoma cells. (**A**) Fold change of ROS generation through various concentrations of H_2_O_2_ stimulation for 1 h on Hep3B–DNA and Hep3B-HO1 cells detected by DCF-DA dye. (**B**) The flow cytometry determined that overexpression of HO-1 blocked ROS generation stimulated by 20 μM of pyocyanin. (**, *p* < 0.01).

**Figure 3 antioxidants-09-00251-f003:**
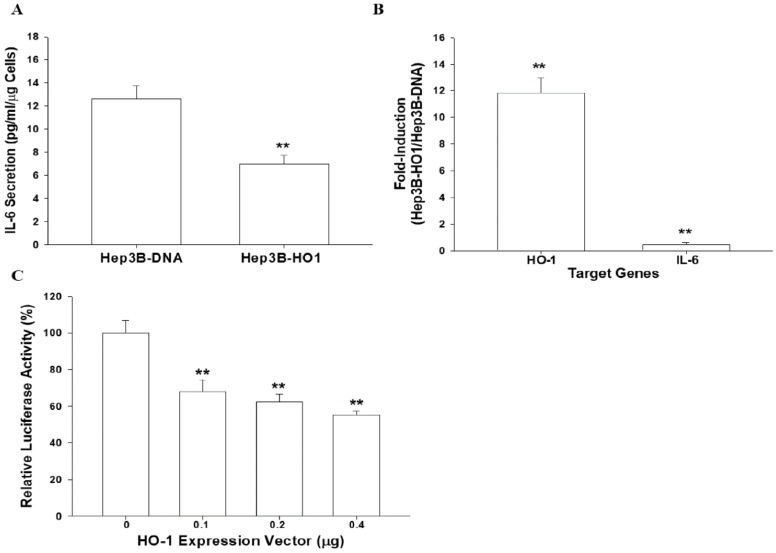
Ectopic overexpression of heme oxygenase-1 blocks interleukin-6 expression. The expression of HO-1 and IL-6 in Hep3B–DNA and Hep3B-HO1 cells were determined by ELISA (**A**), RT-qPCR (**B**) and reporter assays (**C**). Data are expressed as mean ± SE (*n* = 6) of luciferase activity relative to the mock-transfected group. (**, *p* < 0.01).

**Figure 4 antioxidants-09-00251-f004:**
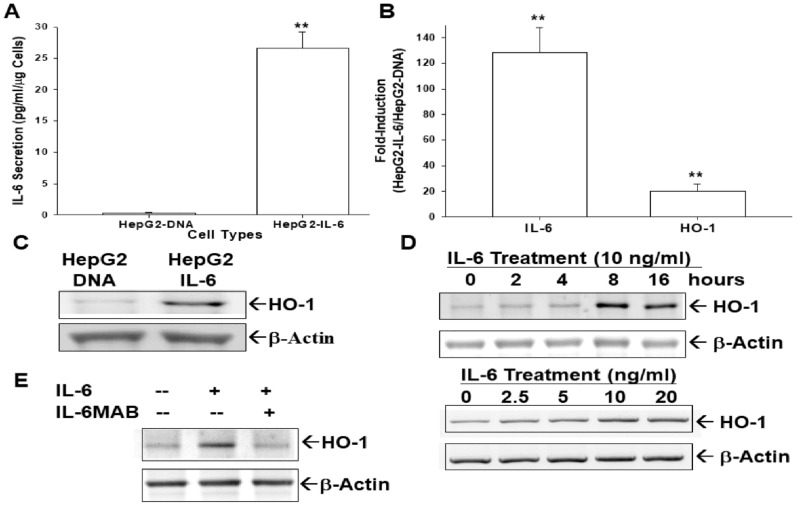
Interleukin-6 induces human heme oxygenase-1 gene expression. HepG2 cells were transiently transfected with pcDNA3zeo-IL-6 (HepG2-IL-6) or pcDNA3.1zeo (HepG2–DNA) expression vectors. The expressions of HO-1 and IL-6 were analyzed using ELISA (**A**), RT-qPCR (**B**), and immunoblot (**C**) assays. (**D**) HepG2 cells were treated with recombinant human IL-6 (10 ng/mL) at various times (top), or with various concentrations of IL-6 as indicated for 16 h (bottom). (**E**) HepG2 cells were treated with IL-6 (10 ng/mL), or IL-6 co-treated with/without an anti-human IL-6 monoclonal antibody (IL-6MAB) for 16 h. Expressions of HO-1 protein levels were analyzed using immunoblot assays. (**, *p* < 0.01).

**Figure 5 antioxidants-09-00251-f005:**
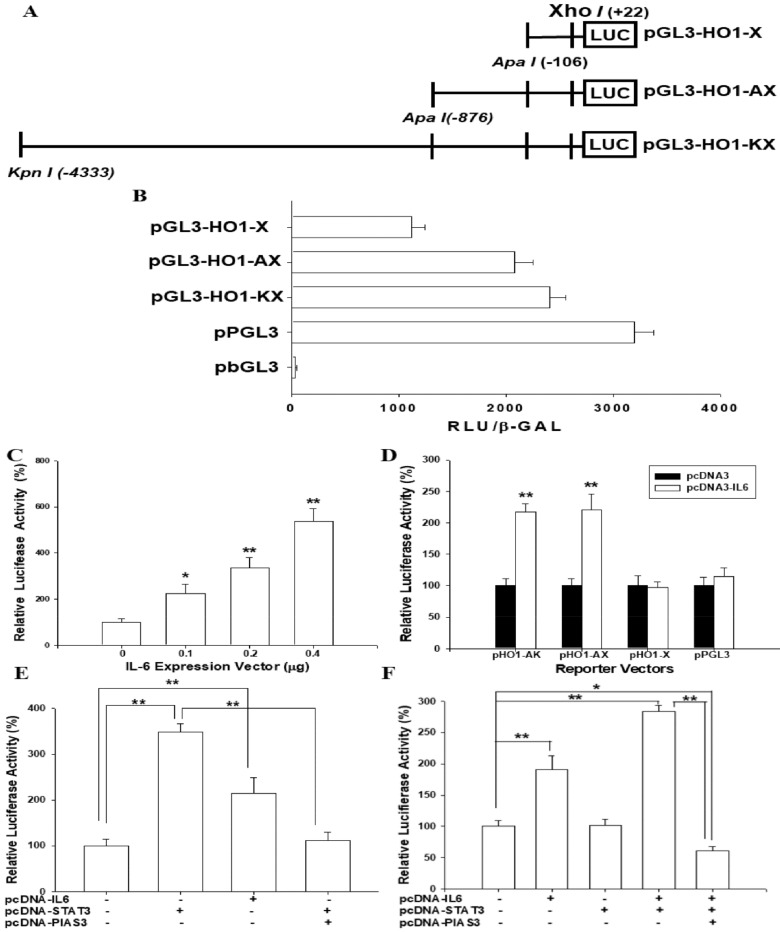
Activation of Interleukin-6 on reporter activity of human heme oxygenase-1 gene is dependent on STAT3 expression. (**A**) Nested deletion constructs of heme oxygenase-1 reporter vectors. (**B**) The reporter activities of the HO-1 reporter vectors and the reporter vectors containing with (ppGL3) or without (pbGL3) SV40 promoter. (**C**) The HO-1 reporter vector (pHO1-KX) was co-transfected with different concentrations of the IL-6 expression vector into HepG2 cells. (**D**) The relative luciferase activities of reporter vectors, containing different fragments from the HO-1 enhancer/promoter, were co-transfected with the IL-6 expression vector (0.2 μg/well) into HepG2 cells. (**E**) The relative luciferase activities of STAT3 reporter vector-transfected HepG2 cells were co-transfected with IL-6 (pcDNA-IL-6), STAT3 (pcDNA-STAT3) and/or PIAS3 (pcDNA-PIAS3) expression vectors. (**F**) The HO-1 reporter vector (pHO1-KX) was transiently co-transfected with IL-6 expression, STAT3 expression and/or PIAS3 expression vectors into HepG2 cells. Data are expressed as mean ± SE (*n* = 6) of luciferase activity relative to the control-treated group or to the group as indicated. (*, *p* < 0.05; **, *p* < 0.01).

**Figure 6 antioxidants-09-00251-f006:**
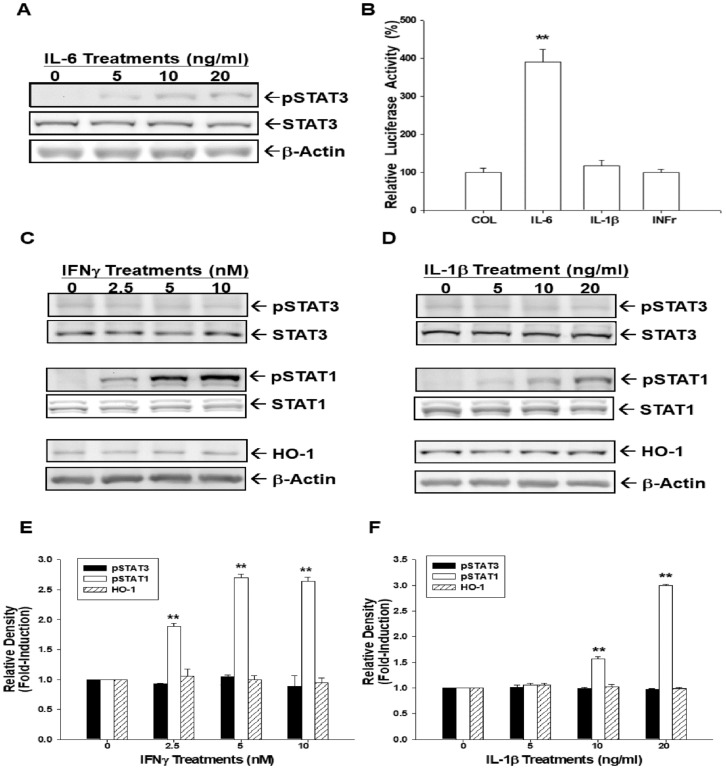
Interleukin-6 induces human heme oxygenase-1 expression via STAT3 phosphorylation. (**A**) HepG2 cells were treated with various concentrations of IL-6, as indicated, for 24 h. Expressions of STAT3 and phospho-STAT3 were determined by immunoblotting. (**B**) The HO-1 reporter vector (pHO1-KX) transfected HepG2 cells were treated with IL-6 (10 ng/mL), IL-1β ng/mL), and interferon γ (10 nM) for 24 h. Data are expressed as mean ± SE (*n* = 6) of luciferase activity relative to the control-treated group. HepG2 cells were treated with various doses of recombinant human INFγ (**C**) and IL-1β (**D**). Protein levels of HO-1, STAT3, phospho-STAT3, STAT1, phospho-STAT1, and β-Actin were analyzed using immunoblot assays. The quantitative data of INFγ (**E**) and IL-1β (**F**) treatments were expressed as the intensity of the protein band of the target gene/β-Actin or phosphorylation protein/total protein relative to the control solvent-treated group (*n* = 3). (**, *p* < 0.01).

**Figure 7 antioxidants-09-00251-f007:**
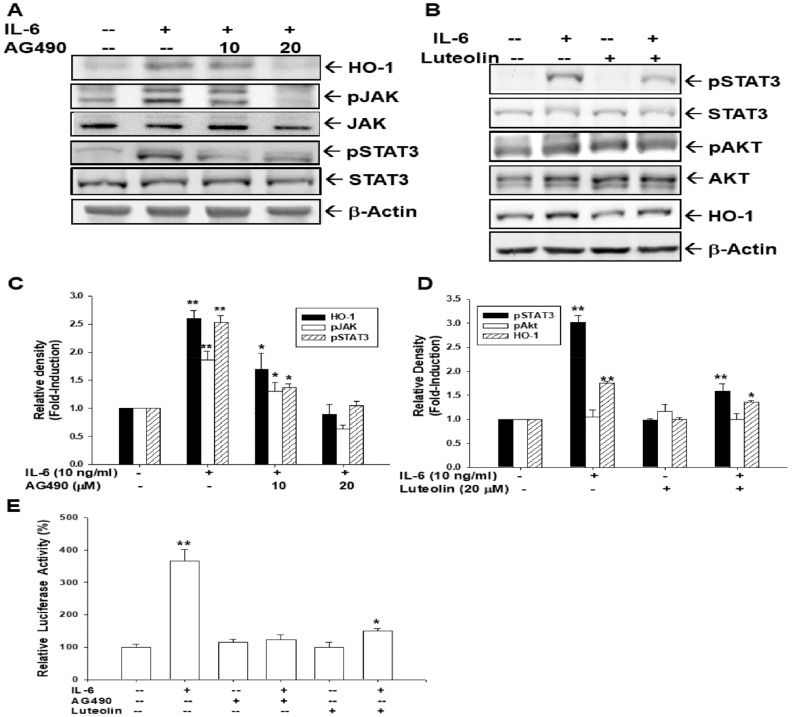
AG490 and luteolin block the activation of Interleukin-6 in heme oxygenase-1 expression through the downregulation of STAT3 phosphorylation. (**A**) HepG2 cells were co-treated by recombinant human IL-6 (10 ng/mL) and various doses of AG490, as indicated, for 16 h. The expression of HO-1, STAT3, and phospho-STAT3, JAK, phospho-JAK, and β-Actin were determined by immunoblot assays. (**B**) HepG2 cells were co-treated with recombinant human IL-6 (10 ng/mL) with or without luteolin (20 μM) for 16 h. Protein levels of HO-1, STAT3, phospho-STAT3, AKT, phospho-AKT, or β-Actin were analyzed using immunoblot assays. The quantitative data of IL-6 and/or AG490 (**C**) and IL-6 and/or luteolin (**D**) treatments were expressed as the intensity of the protein band of the target gene/β-actin or phosphorylation protein/total protein relative to the control solvent-treated group (*n* = 3). (**E**) The HO-1 reporter vector (pHO1-KX) of transfected HepG2 cells was treated with IL-6 (10 ng/mL), AG490 (20 μM), and/or luteolin (20 μM) for 16 h. Data are expressed as mean percent ± SE (*n* = 6) of luciferase activity relative to the solvent-treated group. (*, *p* < 0.05; **, *p* < 0.01).

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
