# Peer review of "Human Heme Oxygenase-1 Induced by Interleukin-6 via JAK/STAT3 Pathways Is a Tumor Suppressor Gene in Hepatoma Cells"

_antioxidants, 2020, doi:10.3390/antiox9030251_

Round 1
Reviewer 1 Report
Human Heme Oxygenase-1 Induced by Interleukin-6 via JAK/STAT3 Pathways is a Tumor Suppressor Gene in Hepatoma Cells
The authors show that Human Heme Oxygenase-1 is involved in cancer in hepatoma cells
I suggest the authors to add in the introduction these recent manuscripts regarding the HO-1
Horio T, Morishita E, Mizuno S, Uchino K, Hanamura I, Espinoza JL, Morishima
Y, Kodera Y, Onizuka M, Kashiwase K, Fukuda T, Doki N, Miyamura K, Mori T, Nakao
S, Takami A. Donor Heme Oxygenase-1 Promoter Gene Polymorphism Predicts Survival
after Unrelated Bone Marrow Transplantation for High-Risk Patients. Cancers
(Basel). 2020 Feb 12;12(2).
Wang LY, Huang CS, Chen YH, Chen CC, Chen CC, Chuang CH. Anti-Inflammatory
Effect of Erinacine C on NO Production Through Down-Regulation of NF-κB and
Activation of Nrf2-Mediated HO-1 in BV2 Microglial Cells Treated with LPS.
Molecules. 2019 Sep 12;24(18)
Pratt R, Lakhani HV, Zehra M, Desauguste R, Pillai SS, Sodhi K. Mechanistic
Insight of Na/K-ATPase Signaling and HO-1 into Models of Obesity and Nonalcoholic
Steatohepatitis. Int J Mol Sci. 2019 Dec 21;21(1). pii: E87. doi:
10.3390/ijms21010087. Review. PubMed PMID: 31877680; PubMed Central PMCID:
PMC6982200.
Kwon SH, Lee SR, Park YJ, Ra M, Lee Y, Pang C, Kim KH. Suppression of
6-Hydroxydopamine-Induced Oxidative Stress by Hyperoside Via Activation of
Nrf2/HO-1 Signaling in Dopaminergic Neurons. Int J Mol Sci. 2019 Nov 20;20(23).
pii: E5832.
Gáll T, Balla G, Balla J. Heme, Heme Oxygenase, and Endoplasmic Reticulum
Stress-A New Insight into the Pathophysiology of Vascular Diseases. Int J Mol
Sci. 2019 Jul 26;20(15). pii: E3675.
Kishimoto Y, Kondo K, Momiyama Y. The Protective Role of Heme Oxygenase-1 in
Atherosclerotic Diseases. Int J Mol Sci. 2019 Jul 24;20(15). pii: E3628.
Lin CC, Hsiao LD, Cho RL, Yang CM. Carbon Monoxide Releasing
Molecule-2-Upregulated ROS-Dependent Heme Oxygenase-1 Axis Suppresses
Lipopolysaccharide-Induced Airway Inflammation. Int J Mol Sci. 2019 Jun
28;20(13). pii: E3157.
Lakhani HV, Zehra M, Pillai SS, Puri N, Shapiro JI, Abraham NG, Sodhi K.
Beneficial Role of HO-1-SIRT1 Axis in Attenuating Angiotensin II-Induced
Adipocyte Dysfunction. Int J Mol Sci. 2019 Jun 29;20(13). pii: E3205.
Podkalicka P, Mucha O, Kruczek S, Biela A, Andrysiak K, Stępniewski J,
Mikulski M, Gałęzowski M, Sitarz K, Brzózka K, Józkowicz A, Dulak J, Łoboda A.
Synthetically Lethal Interactions of Heme Oxygenase-1 and Fumarate Hydratase
Genes. Biomolecules. 2020 Jan 16;10(1).
The dose of luteolin based on what was chosen? The dosage is toxic
The manuscript would benefit from inclusion of introducing/bridging sentences between the individual parts of the "Results" that explain the logical order and rationale for the experiments
In the conclusions , the Authors should highlight the possible clinical significance of their findings
Author Response
For reviewer 1:
- In this manuscript, we found that ectopic overexpression of HO-1 attenuated cell proliferation, invasion, tumorigenesis, and ROS-induction by H2O2 of hepatoma cells. The major issues that we focus are to determined the potential crosstalk between human heme oxygenase-1 (HO-1) and IL-6, and elucidated the signaling pathways involved in the induction of HO-1 by IL-6 in human hepatoma cells. Our results demonstrate that HO-1 is the antitumor gene induced by IL-6 through the IL-6/JAK/STAT3 pathways; moreover, a feedback circuit may exist between IL-6 and HO-1 in hepatoma cells.
- Thanks for your kindly support our manuscript. Grammatical and writing style errors in the original version have been corrected by our colleague who is a native English speaker. We used red font to highlight the modifications in the revised manuscript to indicate where the changes we have made occur in the text. According to the reviewer’s recommendation, several modifications of this manuscript are listed as following
- We agreed the reviewer’s suggestion to added new citations in the section of Introduction to emphasize the biologic function of HO-1 in the recent studies. We added new sentences or modified the sentences in the section of Introduction (citation 3-8, 16) and Discussion (citation 54). Eight new references were added in the section of References.
L43-52: In addition, recent studies have shown that HO-1 involved in the improvements of several events like oxidation stress, airway inflammation, functions of adipocytes, obesity, steatohepatitis, atherosclerosis, diabetes mellitus, brain hemorrhage, as well as neuroprotection [3-8]. This is especially essential in applying to the health maintenance of liver which containing about 15% of body heme [9-12]. Numbers of studies have proved that anti-inflammatory, antioxidant, pro-angiogenic, anti-thrombotic, and anti-apoptotic effects of HO-1 are derived from its catabolic byproducts, CO and biliverdin, from which render HO-1 as a tissue protector and a good candidate for investigating new therapeutic interventions [1,8,13]. However, several researches have extensively demonstrated the positive role of HO-1 in cancer progress [14-16].
L3363-365: Results of this study are similar with other previous studies which demonstrated that HO-1 downregulated IL-6 expression in BV2 microglial cells, macrophages RAW264.7 and prostate carcinoma PC-3 cells [54-56].
L440-454:
- Kwon, S.H.; Lee, S.R.; Park, Y.J.; Ra, M.; Lee, Y.; Pang, C.; Kim, K.H. Suppression of 6-hydroxydopamine-induced oxidative by hyperoside via activation of Nrf2/HO-1 signaling in dopaminergic neurons. J. Mol. Sci.2019, 20, 5832.
- Lin, C.C.;Hsiao, L.D.; Cho, R,L.; Yang, C.M. Carbon monoxide releasing molecule-2-upregulated ROS-dependent heme oxygenase-1 axis suppresses lipopolysaccharide-induced airway inflammation. J. Mol. Sci. 2019, 20, 3157.
- Lakhani, H.V.; Zehra, M.; Pillai, S.S.; Puri, N.; Shapiro, J.I.; Abraham, N.; Sodhi, K. Beneficial role of HO-1-SIRT1 axis in attenuating angiotensin II-induced adipocyte dysfunction. Int. J. Mol. Sci.2019, 20, 3205.
- Pratt, R.;Lakhani, H.V.; Zehra, M.; Desauguste, R.; Pillai, S.S.; Sodhi, K. Mechanistic insight of Na/K-ATPase signaling and HO-1 into models of obesity and nonalcoholic steatohepatitis. J. Mol. Sci. 2019, 21, 87.
- Gáll, T.;Balla, G.; Balla, J. Heme, heme oxygenase, and endoplasmic reticulum stress-A new insight into the pathophysiology of vascular diseases. J. Mol. Sci. 2019, 20, 3675.
- Kishimoto, Y.; Kondo, K.; Momiyama, Y.; The protective role of heme oxygenase-1 in atherosclerotic diseases. J. Mol. Sci. 2019, 20, 3628.
L471-472
- Podkalicka, P.; Mucha, O.; Kruczek, S.; Biela, A.; Andrysiak, K.; Stępniewski, J.; Mikulski, M.; Gałęzowski, M.; Sitarz, K.; Brzózka, K.; et al. Synthetically lethal interactions of heme oxygenase-1 and Fumarate hydratase genes. Biomolecules 2020, 10, 143.
L569-570:
- Wang, L.Y.; Huang, C.S.; Chen, Y.H.; Chen, C.C.; Chen, C.C.; Chuang, C.H. Anti-inflammatory effect of erinacine C on NO production through down-regulation of NF-kB and activation of Nrf2-mediated HO-1 in BV2 microglial cells treated with LPS. Molecules 2019, 24, 3317.
- The reason why we used the 20 mM of luteolin in this study is based on our previous study (Tsui KH, Chung LC, Feng TH, Chang PL, Juang HH. 2012. Upregulation of prostate-derived ets factor by luteolin causes inhibition of cell proliferation and cell invasion in prostate carcinoma cells. International Journal of Cancer 130: 2812-2823). In that study, we found when treatments of luteolin > 30 mM caused the cytotoxicity effect on prostate carcinoma cells. Therefore, we used 20 mM luteolin in this study since this dosage was found less cytotoxicity in most studies.
- We agreed the reviewer’s suggestion to add the description of the approach and rationale of the results. We added or remodified several sentences in each sub-section of Results as following
L178-179: In order to determine the biologic functions of HO-1 in hepatoma cells, we overexpressed ectopic HO-1 into the cells.
L200-201: We continued to determine the anti-proliferation activity of HO-1 in vivo by using xenograft animal studies.
L210-211: We determined the antioxidant response of HO-1 in human Hep3B cells. Results of the total ROS assays indicated that
L224-225: In order to determine the anti-inflammatory characteristics of HO-1 in hepatoma cells, we evaluated the effect of HO-1 on IL-6 gene expression.
L239-249: We investigated the crosstalk between IL-6 and HO-1 in hepatoma cells since IL-6 is known as one of the HO-1 stimulator.
L260: We continued to investigate the gene regulation of HO-1 by IL-6 in hepatoma cells.
L293-294: We continued to determine the specific STAT3 signal pathways in the effect of IL-6 on HO-1 gene expression.
L319-320: Taken together, these results proved that IL-6 affects HO-1 expression through a specific JAK/STAT3 signaling.
- We agreed the reviewer’s suggestion to highlight the possible clinical significance in the section of Conclusions
L423-424: Further studies of application of HO-1 and its inducers against tumorigenesis and inflammatory effects in hepatocellular carcinoma are warranted.

Reviewer 2 Report
The authors clearly demonstrated that hepatoma cells were suppressed by HO-1 induced by IL-6 via JAK/STAT pathway. This paper is interesting and well written. However, there are several issues for acceptance.
1. Previous report (Basu C, et al. 2018 Biomed Res Int) demonstrated that ROS induced by H2O2 promoted apoptosis in HepG2 cells; however, in this study, overexpression of HO-1 hepatoma cells decreased cell proliferation, even though ROS induced by H2O2 was alleviated. Dose the alleviation of ROS mediated by HO-1 contribute to the suppression of hepatoma cell proliferation and invasion? Did H2O2 stimulation in Hep3B-HO1 attenuate cell viability compared to Hep3B-DNA? The authors should discuss the association between the alleviation of H2O2-induced ROS and suppression of hepatoma cells.
2. As the authors mentioned, some studies demonstrated that HO-1 expression contributed to progression of hepatoma, such as anti-apoptosis (Tanaka S, et al. 2003 Br J Cancer) in hepatoma cell lines, and microvascular and capsular invasion in clinical subjects (Park CS, et al. 2019 Medicine). Why were the results discrepancy in the previous studies? The authors should discuss these points.
3. What did the asterisks in Figure 3B and 4B compared to? This point is unclear.
Author Response
For reviewer 2:
- In this manuscript, we found that ectopic overexpression of HO-1 attenuated cell proliferation, invasion, tumorigenesis, and ROS-induction by H2O2 of hepatoma cells. The major issues that we focus are to determined the potential crosstalk between human heme oxygenase-1 (HO-1) and IL-6, and elucidated the signaling pathways involved in the induction of HO-1 by IL-6 in human hepatoma cells. Our results demonstrate that HO-1 is the antitumor gene induced by IL-6 through the IL-6/JAK/STAT3 pathways; moreover, a feedback circuit may exist between IL-6 and HO-1 in hepatoma cells.
- Thanks for your kindly support our manuscript. Grammatical and writing style errors in the original version have been corrected by our colleague who is a native English speaker. We used red font to highlight the modifications in the revised manuscript to indicate where the changes we have made occur in the text. According to the reviewer’s recommendation, several modifications of this manuscript are listed as following
- The reason why we did the experiments of Figure 2 is that we wanted to evaluate ectopic overexpression of HO-1 has the antioxidant characteristics induced by H2O2. That is to confirm the ectopic-overexpressed HO-1 still has the biologic function. We did not compare the cell viability between mock-transfected and HO-1 transfected hepatoma cells. However, we agreed the reviewer’s suggestions. We remodified and added the sentences in the section of Discussions to emphasize the correlation among ROS, cell viability, proliferation , and invasion as following.
L355-360: These results supported recent studies suggesting that upregulation of HO-1 alleviated the H2O2-induced ROS and oxidative injury in HepG2 cells [50,51] although, in this study, we did not determine whether ectopic HO-1 overexpression inhibited the apoptosis of hepatoma cells. Since recent study indicated that IL-6-driven ROS reduction by increasing Nrf 2, a transcription activator of HIO-1 gene, expression in human islet b-cells [52], further investigation in the role of IL-6-HO-1 crosstalk in hepatoma cells in terms of ROS synthesis is warranted.
- We agreed the reviewer’s concerning the divergent results of the function of HO-1 in tumor biology. We remodified and added several sentences as the reviewer’s suggestions in the section of Discussions
L344-353: The investigations of the role of HO-1 with contrary results have been extensively discussed in the studies of liver carcinogenesis in vitro or in vivo [42-48]. Studies of immunohistochemical staining showed that HO-1 is frequently overexpressed in human hepatocellular carcinoma (HCC); however, HO-1 expression did not seem to influence the prognosis of HCC [42,43]. Studies using the rat hepatoma AH1368 cells and mouse hepatoma cells, Hepa129 and Hepa1-6, indicated that HO-1 is regarded as an oncogene in orthotopic tumor assays [42,44]. However, results from other studies showed that overexpression of HO-1 in HepG2 retarded cell proliferation, migration, invasion, and orthotopic tumor growth which are in agreement with our in vitro or in vivo studies using human hepatoma cells [47,48]. The mechanisms of HO-1 in the attenuation of HCC progression through microRNA pathway has been discussed [49].
- Four references as reviewer suggested were also added in the section of References.
L539-543:
- Park, C.S.; Eom, D.W.; Ahn, Y.; Jang H.J.; Hwang, S.; Lee, S.G. Can heme oxygenase-1 be a prognostic factor in patients with hepatocellular carcinoma? Medicine 2019, 98, 16084.
- Tanaka, S.; Akaike, T.; Fang, J.; Beppu, T.; Ogawa, M.; Tamura, F.; Miyamoto, Y.; Maeda, H. Antiapoptotic effect of haem oxygenase-1 induced by nitric oxide in experimental solid tumour. J. Cancer 2003, 88, 902-909.
L555-557:
- Zou, C.; Zou, C.; Cheng, W.; Li, Q.; Han, A.; Wang, X.; Jin, J.; Zou, J.; Liu, Z, Zhou, Z.; et al. Heme oxygenase-1 retards hepatocellular carcinoma progression through the microRNA pathway. Oncol Rep. 2016, 36, 2715-2722.
L564-566:
- Marasco, M.R.; Contech, A.M.; Reissaus, C.A.; Cupit V, J.E.; Appleman, E.M.; Mirmira, R.G.; Linnemann, A.K. Interleukin-6 reduces b-cell oxidative stress by linking autophagy with the antioxidant response. Diabetes 2018, 67, 1675-1588.
- The asteristks in Figure 3B and 4B represented the comparison between Hep3B-DNA and HepG3-HO1, and HepG2-DNA and HepG2-IL-6, respectively. We has showed in the legend of x-axis in each figure. However, we agreed reviewer’s concerning and remodified the sentences in the section of Results as following.
L227-230: The quantitative results of RT-qPCR assays revealed that ectopic HO-1 overexpression enhanced 12.08 ± 1.88 folds and 0.41 ± 0.10 folds, respectively, of HO-1 and IL-6 mRNA levels in Hep3B cells, (Figure 3B) indicating that ectopic overexpression of HO-1 downregulated gene expression of IL-6 as compared with Hep3B-DNA cells.
L242-245: The quantitative results of RT-qPCR assays revealed that ectopic IL-6 overexpression enhanced 128.1 ± 19.9 folds and 20.1 ± 5.3 folds, respectively, of IL-6 and HO-1 mRNA levels indicating that ectopic overexpression of IL-6 induced gene expression of HO-1 in HepG2 cells (Figure 4B).
Reviewer 3 Report
The authors have presented an interesting work and demonstrate that HO-1 is the antitumor gene induced by IL-6 through the IL-6/JAK/STAT3 pathways. In addition, a feedback circuit may exist between IL-6 and HO-1 in hepatoma cells. However, I have some questions that I would like to be answered.
Lines 206-211. The relationship between HO-1 and the Ros generation is not very well explained.
Lines 338-340. Why the effectof HO-1 on IL-6 expression might be dependen ton cell types?
The anti-inflammatory effect of HO-1 is due to a negative feedback loop between HO-1 and IL-6?
What role does IL-6 play on ROS synthesis?
Author Response
For reviewer 3:
- In this manuscript, we found that ectopic overexpression of HO-1 attenuated cell proliferation, invasion, tumorigenesis, and ROS-induction by H2O2 of hepatoma cells. The major issues that we focus are to determined the potential crosstalk between human heme oxygenase-1 (HO-1) and IL-6, and elucidated the signaling pathways involved in the induction of HO-1 by IL-6 in human hepatoma cells. Our results demonstrate that HO-1 is the antitumor gene induced by IL-6 through the IL-6/JAK/STAT3 pathways; moreover, a feedback circuit may exist between IL-6 and HO-1 in hepatoma cells.
- Thanks for your kindly support our manuscript. Grammatical and writing style errors in the original version have been corrected by our colleague who is a native English speaker. We used red font to highlight the modifications in the revised manuscript to indicate where the changes we have made occur in the text. According to the reviewer’s recommendation, several modifications of this manuscript are listed as following
- The reason why we did the experiments of Figure 2 is that we wanted to evaluate ectopic overexpression of HO-1 has the antioxidant characteristics induced by H2O2. That is to confirm the ectopic-overexpressed HO-1 still has the biologic function. We agreed the reviewer’s concerning about the relationship between HO-1 and ROS. We remodified and added the sentences in the section of Results and Discussions to emphasize the correlation between ROS and HO-1 in this manuscript as following.
L214-217: Further flow cytometry assays indicated that ROS generation was stimulated by 20 mM of pyocyanin, a ROS inducer, in Hep3B-DNA cells; however, ectopic HO-1 overexpression blocked the pyocyanin-stimulated ROS generation in Hep3B-HO1 cell as compared with Hep3B-DNA cells (Figure 2B).
L353- 357: Moreover, as shown in figure 2, our results also demonstrated that overexpression of HO-1 in Hep3B cells attenuated H2O2-induced ROS generation. These results supported recent studies suggesting that upregulation of HO-1 alleviated the H2O2-induced ROS and oxidative injury in HepG2 cells [50,51] although, in this study, we did not determine whether ectopic HO-1 overexpression inhibited the apoptosis of hepatoma cells.These results supported recent studies suggesting that upregulation of HO-1 alleviated the H2O2-induced ROS and oxidative injury in HepG2 cells [50,51] although, in this study, we did not determine whether ectopic HO-1 overexpression inhibited the apoptosis of hepatoma cells.
- We agreed the reviewer’s concerning why the effect of HO-1 on IL-6 might be dependent on cell types. We remodified the sentences as in the section of Discussions
L363-367: Results of this study are similar with other previous studies which demonstrated that HO-1 downregulated IL-6 expression in BV2 microglial cells, macrophages RAW264.7 and prostate carcinoma PC-3 cells [54-56]. However, the effect of HO-1 on IL-6 expression might be dependent on cell types since study of multiple myeloma cells showed contrary results in which HO-1 induced IL-6 expression [23]
- We agreed reviewer’s concerning in the explain the effect of the negative regulation between IL-6 and HO-1 on the anti-inflammatory We added one sentence in the section of Discussion as following
L381-382: and the negative feedback between HO-1 and IL-6 might be due to partly the anti-inflammatory effect of HO-1 [14].
- We agreed reviewer’s concerning in the role of IL-6 on ROS synthesis although this was not the issue for this manuscript. We added one sentence in the section of Discussion as following. One new reference also was added in the section of References
L357-360: Since recent study indicated that IL-6-driven ROS reduction by increasing Nrf 2, a transcription activator of HO-1 gene, expression in human islet b-cells [52], further investigation in the role of IL-6-HO-1 crosstalk in hepatoma cells in terms of ROS synthesis is warranted.
L564-566:
- Marasco, M.R.; Contech, A.M.; Reissaus, C.A.; Cupit V, J.E.; Appleman, E.M.; Mirmira, R.G.; Linnemann, A.K. Interleukin-6 reduces b-cell oxidative stress by linking autophagy with the antioxidant response. Diabetes 2018, 67, 1675-1588.
Round 2
Reviewer 2 Report
This manuscript is well revised according to the reviewer's comments.
Reviewer 3 Report
The authors have significantly improved the work and have adequately answered the questions asked.